# Sound suppresses earliest visual cortical processing after sight recovery in congenitally blind humans

Suddha Sourav [1✉], Ramesh Kekunnaya [2], Davide Bottari [1,3], Idris Shareef [2], Kabilan Pitchaimuthu[1,2,4] & Brigitte Röder[1,2]

Neuroscientific research has consistently shown more extensive non-visual activity in the visual cortex of congenitally blind humans compared to sighted controls; a phenomenon known as crossmodal plasticity. Whether or not crossmodal activation of the visual cortex retracts if sight can be restored is still unknown. The present study, involving a rare group of sight-recovery individuals who were born pattern vision blind, employed visual event-related potentials to investigate persisting crossmodal modulation of the initial visual cortical processing stages. Here we report that the earliest, stimulus-driven retinotopic visual cortical activity (<100 ms) was suppressed in a spatially specific manner in sight-recovery individuals when concomitant sounds accompanied visual stimulation. In contrast, sounds did not modulate the earliest visual cortical response in two groups of typically sighted controls, nor in a third control group of sight-recovery individuals who had suffered a transient phase of later (rather than congenital) visual impairment. These results provide strong evidence for persisting crossmodal activity in the visual cortex after sight recovery following a period of congenital visual deprivation. Based on the time course of this modulation, we speculate on a role of exuberant crossmodal thalamic input which may arise during a sensitive phase of brain development.

[1] Biological Psychology and Neuropsychology, University of Hamburg, Hamburg, Germany. [2] Jasti V Ramanamma Children's Eye Care Center, Child Sight Institute, L V Prasad Eye Institute, Hyderabad, India. [3] IMT School for Advanced Studies Lucca, Lucca, Italy. [4] Department of Medicine and Optometry, Linnaeus University, Kalmar, Sweden. ✉email: suddha.sourav@uni-hamburg.de

In sensitive periods, the developing brain is characterized by a heightened capacity for shaping its neural circuits to optimally process the available sensory landscape[1]. For example, people born blind have been found to acquire higher skills in processing information from the intact, non-visual sensory systems[2–4]. Such compensatory performance in permanent blindness has been shown to accompany changes in neural systems associated with the intact modalities (e.g. the auditory cortex)[5], as well as crossmodal activations of what is typically considered the visual cortex. In fact, crossmodal activation of both striate (primary) and extrastriate (non-primary) regions of blind individuals' visual cortex, induced by auditory and tactile stimuli, is a consistent finding in neuroimaging and electrophysiological investigations not only in humans but across numerous mammalian species, including rodents, cats, and non-human primates[6–11].

At the same time, individuals who were born without pattern vision, but regained sight later in life, typically feature severe visual impairments. In children born pattern vision blind due to the presence of bilateral, dense congenital cataracts, each week of delay in surgery reduces the visual acuity attained afterwards[12]. Additionally, patients treated for congenital cataracts have been shown to have compromised visual acuity as well as degraded mid- to higher-level vision[13,14], and impairments in some audiovisual functions[15]. The consequences of an equal period of visual deprivation are comparatively less serious later in childhood[16,17], and full recovery has been often observed after a transient phase of adult blindness[18].

In permanently congenitally blind humans, two main routes have been proposed for the crossmodal (i.e., non-visual) activity observed in the visual cortex. First, a higher innervation of the visual cortex by thalamocortical projections from non-visual nuclei has been suggested[11]. This hypothesis is compatible with anatomical evidence from anophthalmic or enucleated non-human mammals indicating direct thalamocortical projections to the visual cortex from multiple nonvisual thalamic nuclei[19,20]. Second, cortico-cortical routes have been proposed as a parsimonious alternative[21,22]. This idea has been supported by evidence for the existence of fast, direct cortico-cortical connections between primary sensory areas in typically developing mammals which could provide an anatomical pathway for crossmodal activation of the visual cortex in blindness[23,24]. However, whether crossmodal activity of early visual cortex persists or retracts after sight restoration and possibly interferes with visual recovery is yet widely unknown. For example, non-visual representations could hypothetically emerge during a sensitive period and thus might permanently occupy synaptic space in the visual cortex[25,26], resulting in a suppression of visually evoked activity[27]. Yet currently no data exist that would allow answering to what degree, at which level, and in which manner crossmodal information might influence visual cortical activity after sight restoration. Investigations in shortly visually deprived individuals who were born with bilateral cataracts have indicated some auditory evoked activity in extrastriate cortex[28,29] and a suppression of visual evoked activity by concurrent auditory stimulation[30]. A case report of an early, though presumably not congenitally visually deprived individual additionally found striate cortical responses before as well as shortly after sight recovery[31]. However, blood-oxygen-level-dependent (BOLD) signal changes measured by functional magnetic resonance imaging (fMRI), utilized in the preceding studies, unfold over multiple seconds, and thus do not possess the required temporal resolution to untangle the role of feedback activity from higher order regions from bottom-up, stimulus-driven auditory activity as the main source of auditory influence on early visual cortical areas[32]. Importantly, crossmodal activation reported in sight-recovery individuals' visual cortex was modest compared to the widespread crossmodal activity typically observed in permanently blind humans[2,3]. This observation has led to the hypothesis that some but not all routes causing crossmodal activity in permanently blind humans might retract after sight restoration[3,33].

The central aim of the present study was to test whether bottom-up, stimulus-driven auditory activity modulates visual processing after sight recovery. To this end, we exploited the high temporal resolution of electroencephalographic (EEG) recordings (in the order of milliseconds) in a rare, well characterized group of sight-recovery individuals who were born pattern vision blind and later surgically regained their vision[16]. These individuals were presented visual stimuli targeting the opposite banks of the calcarine sulcus (CaS), which houses most of the human V1 (Fig. 1a–c). In some trials, concomitant white noise bursts accompanied the visual stimulus (Fig. 1d). This protocol enabled us to derive the C1 wave of visual event-related potentials (ERPs), which reflects the earliest visual cortical response and typically appears 50–100 ms after stimulus onset[34,35]. The C1 difference wave ($\Delta C1$) was calculated by subtracting the ERPs elicited by upper visual field (UVF) stimulation from those elicited by lower visual field (LVF) stimulation, i.e., $\Delta C1 \overset{\text{def}}{=} v_{LVF} - v_{UVF}$. This procedure allowed the separation of retinotopic activity from any unspecific activity (e.g., non-retinotopic neural activity as well as neural activities that are common to both the UVF and the LVF stimuli), and thus indexes the genuine retinotopic activity in early visual cortex, with V1 likely the strongest contributor[36,37]. Any difference between the $\Delta C1$ between unimodal visual vs. crossmodal (audiovisual) conditions thus reflects sound evoked changes in bottom-up activity in early visual cortex, especially V1[37].

Humans react faster to concurrent crossmodal, e.g., audiovisual stimuli than to any of the constituent (i.e., auditory or visual) stimuli presented alone[38–41]. Despite persisting visual processing deficits as well as impairments in some higher-level audiovisual tasks, e.g., audiovisual speech fusion[15,42], some other multisensory functions have been reported to be spared after congenital cataract reversal, including faster responses to crossmodal compared to unimodal stimuli[43–45]. Yet, whether bottom-up integration of auditory and visual information in V1 is critical for reaction time improvements is a central unanswered question in multisensory research. The second goal of the present study was to ascertain whether faster reaction times due to concurrent audiovisual stimulation requires the integration of auditory and visual information at the earliest retinotopic visual processing stage.

Here we demonstrate a modulation of the $\Delta C1$ by concomitant sounds only in individuals with reversed congenital cataracts, but not in any of the control groups. The present time-resolved analysis thus provides unambiguous evidence for persisting crossmodal stimulus-driven activity interfering with feedforward visual cortical processing even many years after sight restoration in individuals born pattern vision blind. In contrast, all participant groups showed faster reaction times to audiovisual than to unimodal visual stimulation, indicating that the integration of bottom-up auditory and visual information in the earliest retinotopic processing stage in V1 is not a prerequisite for multisensory reaction time improvements involving simple audiovisual stimuli.

## Results

Fourteen sight recovered participants with a history of total, dense, and bilateral congenital cataracts took part in the experiment (CC group, mean age = 17.07 years, range = 6–39 years). The CC participants were not able to perceive more than diffuse light through the cataracts before their surgery (mean duration of

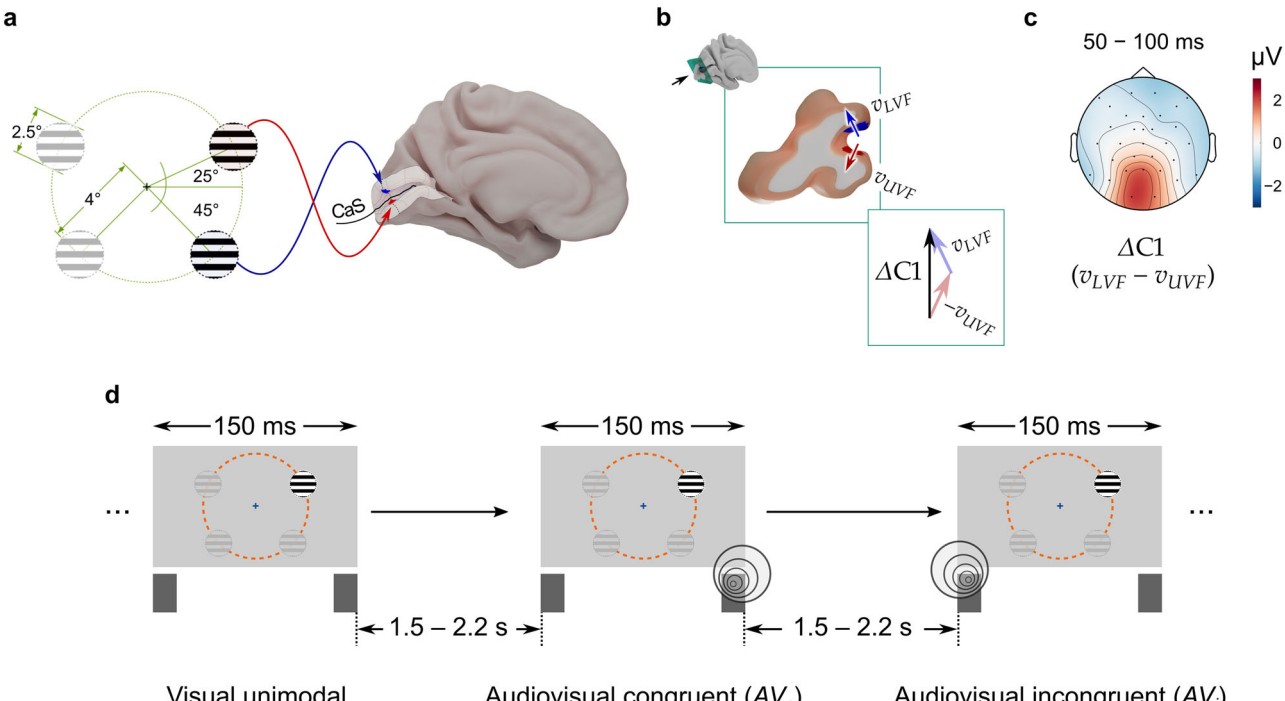

**Fig. 1 Experimental paradigm and trial structure. a** Visual stimuli were presented one at a time in one of the four visual field quadrants. The locations were chosen to target opposite banks of the calcarine sulcus (CaS)[35,127] where most of the human primary visual cortex (V1) is located. **b** Equivalent current dipoles in V1 for the stimulated locations in the upper (UVF) and the lower (LVF) visual field. Subtracting upper visual field potentials ($v_{UVF}$) from lower visual field potentials ($v_{LVF}$) produces the C1 difference wave ($\Delta C1$), emphasizing retinotopic activity while eliminating non-retinotopic and/or common neural activity[36,47]. **c** Mean scalp topography of the $\Delta C1$ in typically sighted controls, 50–100 ms after stimulus onset ($n = 29$). The topography, sign, and latency of the $\Delta C1$ is consistent with a V1 origin[34,35,52,128]. **d** Trial structure. Participants saw brief circular grating patterns in one of the four quadrants or heard white noise bursts (150 ms) from the left or the right loudspeaker. The auditory and the visual stimuli were either presented alone (unimodal, e.g., left diagram) or together (audiovisual, AV). In AV trials, auditory and visual stimuli appeared either from the same side (spatially congruent, e.g., middle diagram), or opposite sides (spatially incongruent, e.g., right diagram). Participants detected rarely (20%) presented vertical oriented gratings, or a deviant sound of the same duration (not pictured). A uniformly distributed interstimulus interval between 1.5–2.2 s followed each stimulus.

visual deprivation = 42.14 months, range = 1 month – 17.75 years). They were tested at least three years after undergoing surgery (see *Methods: Participants*), guaranteeing an extended time for visual recovery (geometric mean visual acuity at test = 0.229, decimal units, range = 0.051–0.7). Fifteen additional participants with a history of bilateral developmental cataracts (DC group), who underwent the same surgical procedures, were included as a control group (mean age = 14.47 years, range = 9–24 years). This group was considered to control for surgery-related factors (e.g., seeing with intraocular lenses) and the role of vision after birth. For each CC and DC individual, a typically sighted participant matched for age, sex, and handedness took part (MCC: matched controls for the CC group, $n = 14$, MDC: Matched controls for the DC group, $n = 15$). The CC and DC groups were tested at the L V Prasad Eye Institute in Hyderabad, India, and the typically sighted control participants were tested in Hamburg, Germany (see *Methods: Participants*).

In the experiment, participants were exposed to unimodal (i.e., auditory or visual), as well as bimodal (audiovisual) stimuli. The visual stimuli were circular grating patterns appearing one at a time in one of the four visual field quadrants for 150 ms (Fig. 1a and *Methods*). Auditory stimuli consisted of an approximate white noise burst, also 150 ms long, played from one of two loudspeakers located either to the left or to the right side of the screen. Bimodal stimuli were the eight possible combinations of a simultaneously occurring auditory and a visual stimulus (2 sides, left/right, for auditory stimuli × 4 visual field quadrants for visual stimuli). Twenty percent of the stimuli comprised either a rare

visual stimulus (vertical grating orientations, instead of the more commonly presented horizontal orientation) or a rare auditory stimulus (approximate white noise bursts with interruptions, see *Methods*) or both. The standard stimuli ($P = 80\%$) did not require any response. Participants had to respond to the rare visual or auditory stimuli (i.e., targets, $P = 20\%$). In all groups, the hit rate was above 90%, and the false positive rate below 2.5% (in unimodal visual condition: hit rates > 85%, false positive rates < 2.5%).

We analyzed only ERPs elicited by the standard ($P = 80\%$) stimuli to ensure a high signal-to-noise ratio and to avoid a potential confound with motor responses accompanying the target stimuli. Reaction time data analyses were based on the target stimuli (*Methods*). For all visual and bimodal stimuli with a left visual field stimulation, we remapped the electrodes offline, mirroring the electrodes with an anterior-posterior axis of reflection, thus doubling the number of trials (see Supplementary Note 1 and Supplementary Table 1)[16,46]. After this remapping procedure, visual stimuli can be thought of as always appearing on the right side, either in the upper visual field (UVF), or in the lower one (LVF). For bimodal stimuli, the additional concurrent sound came either from the same (congruent), or from the opposite (incongruent) side of visual stimulation. The six stimulus conditions after remapping were thus: visual ($V_{UVF}$ and $V_{LVF}$), audiovisual incongruent ($AV_{i,UVF}$ and $AV_{i,LVF}$), and audiovisual congruent ($AV_{c,UVF}$ and $AV_{c,LVF}$). From these six stimuli, three C1 difference waves were derived for each participant: for the $V$, the $AV_i$, and the $AV_c$ condition.

**Table 1 Electrodes with substantial evidence for the presence of a $\Delta C1$.**

| Condition | Group | Electrode | Mean $\Delta C1(\mu V)$ | Credible interval ($CrI$, $\mu V$) | Bayes factor ($BF_{+0}$) |
|---|---|---|---|---|---|
| Visual (V) | CC | O1 | 1.74 | [0.72–2.77] | 50.29 |
| | MCC | O1 | 2.36 | [1.33–3.39] | $1.29 \times 10^3$ |
| | MCC | O2 | 1.69 | [0.66–2.71] | 34.51 |
| | MCC | P3 | 1.43 | [0.40–2.46] | 8.08 |
| | MCC | Pz | 1.85 | [0.81–2.88] | 84.23 |
| | DC | O1 | 2.30 | [1.18–3.41] | 320.38 |
| | MDC | O1 | 1.70 | [0.57–2.83] | 16.82 |
| | MDC | Pz | 1.46 | [0.33–2.59] | 5.18 |
| Audiovisual incongruent ($AV_i$) | CC | – | – | – | – |
| | MCC | O1 | 2.22 | [1.18–3.24] | 945.89 |
| | MCC | O2 | 1.35 | [0.32–2.37] | 5.78 |
| | DC | O1 | 1.50 | [0.38–2.62] | 6.35 |
| | MDC | O1 | 1.84 | [0.72–2.95] | 28.62 |
| | MDC | Pz | 1.52 | [0.40–2.64] | 7.05 |
| Audiovisual congruent ($AV_c$) | CC | – | – | – | – |
| | MCC | O1 | 1.80 | [0.77–2.83] | 65.42 |
| | MCC | P3 | 1.48 | [0.45–2.51] | 10.93 |
| | MCC | Pz | 1.80 | [0.77–2.83] | 68.79 |
| | DC | O1 | 1.86 | [0.73–2.97] | 38.6 |
| | MDC | O1 | 1.47 | [0.35–2.58] | 5.73 |
| | MDC | O2 | 1.54 | [0.41–2.65] | 7.95 |

Only electrodes with substantial evidence for the presence of a $\Delta C1$ are listed (one-sided Bayes factor, $BF_{+0} > 3$, and where the 95% credible interval ($CrI$) fell outside of the region of practical equivalence, ROPE).
CC Congenital cataract reversal individuals, MCC Matched controls for the CC group, DC Developmental cataract reversal individuals, MDC Matched controls for the DC group.

**Unimpaired Visual $\Delta C1$ in Sight-recovery Humans.** The presence of a C1 difference wave ($\Delta C1$, mean over 50–100 ms, Methods) was investigated at five preselected parietal/occipital electrodes (P3, Pz, P4, O1, and O2), where the C1 wave is known to be most prominent[47] (see pilot experiment, Supplementary Note 2 and Supplementary Figs. 1–3). The Bayesian analysis employed two hierarchical models – one for the CC participants and their matched controls (MCC group), and the other for the DC participants and their matched control group (MDC group). The presence of a $\Delta C1$ was ascertained by a one-sided Bayes factor indicating substantial evidence for the presence of a $\Delta C1$ ($BF_{+0} > 3$), combined with the 95% credible interval ($CrI$) of the $\Delta C1$ falling outside a null region of practical equivalence (ROPE test, Methods: ERP Analysis)[48].

In the visual condition, substantial evidence for the presence of a $\Delta C1$ was observed in all groups (Table 1 and Fig. 2). Specifically, all groups exhibited a $\Delta C1$ at the occipital electrode O1. In addition, in the MCC group, substantial evidence for a $\Delta C1$ was also found at the electrodes O2, P3, and Pz, and in the MDC group, at the electrode Pz (Fig. 2). The same two Bayesian hierarchical models, reparametrized with custom contrasts, were used to test a-priori hypotheses regarding $\Delta C1$ differences between conditions and groups[49]. Here we calculated two-sided Bayes factors in the absence of a-priori hypotheses for effect directions (Methods: ERP Analysis). In the visual condition, we did not find any substantial differences between the groups' $\Delta C1$ values.

The existence of a $\Delta C1$ in CC individuals for visual stimulation replicates findings from an independent previous study of our group[46], indicating that after sight restoration, visual processing takes place in a retinotopically organized visual cortex even after extended periods of congenital visual deprivation.

**Auditory stimuli abolishes the $\Delta C1$ in the CC group, but not in controls.** In stark contrast to the unimodal visual condition, substantial evidence for a $\Delta C1$ was not observed in any of the audiovisual conditions in the CC group, i.e., neither for the spatially incongruent ($AV_i$) nor for the congruent ($AV_c$) condition. Crucially, concomitant sounds did not suppress the $\Delta C1$ in any of the three control groups (DC, MCC, and MDC), and substantial evidence for a $\Delta C1$ emerged in these groups for the incongruent ($AV_i$) as well as the congruent ($AV_c$) audiovisual condition (Table 1 and Fig. 2). Using reparametrized Bayesian hierarchical models to investigate differences between unimodal vs. crossmodal $\Delta C1$s within each group, and between-group differences in $\Delta C1$s in each condition, substantial evidence for a reduced $\Delta C1$ for the $AV_c$ condition compared to the unimodal visual condition in the CC group was found (O1, difference estimate [95% $CrI$]: 2.12 [0.91–3.33] $\mu V$, Cohen's $d = 0.82$, $BF_{10} = 36.90$; and O2, 1.6 [0.39–2.81] $\mu V$, $d = 0.76$, $BF_{10} = 3.54$). Additionally, the $\Delta C1$ was substantially lower in the $AV_c$ condition in the CC group compared to the MCC group (Pz, difference estimate [95% $CrI$]: 1.84 [0.40–3.27] $\mu V$, $d = 1.41$, $BF_{10} = 3.39$; and O1, 2.14 [0.71–3.57] $\mu V$, $d = 1.10$, $BF_{10} = 9.95$). No between-group differences emerged for DC/MDC comparisons. Additionally, in the DC group, the $\Delta C1$ amplitude in the $AV_i$ or $AV_c$ conditions did not substantially differ compared to the unimodal visual condition.

In the CC group, qualitatively, both at the grand average level as well as for individual data points, the $\Delta C1$ suppression appeared to be stronger for the spatially congruent ($AV_c$) than for the spatially incongruent audiovisual condition ($AV_i$; see Fig. 2). To examine whether the $\Delta C1$ suppression depends on spatial congruence, we modeled the $\Delta C1$ in the CC group as a function of stimulus condition in an exploratory analysis. An ordered factor with the levels ($V$, $AV_i$, $AV_c$), increasing in spatially specific crossmodal influence, served as the independent variable. The Bayesian analysis indicated very strong evidence for a general linear trend across the five electrodes in the CC group ($V > AV_i > AV_c$; $BF_{10} = 211.03$, positive ROPE test), indicating a $\Delta C1$ suppression modulated by spatial congruence.

In contrast, after combining the MCC and the MDC groups, not even anecdotal evidence of the $\Delta C1$ being different for either the $AV_i$ or the $AV_c$ condition, as compared to the unimodal visual condition, was obtained at any electrode (see Supplementary Note 3, Supplementary Fig. 4, and Supplementary Table 2, maximum $BF_{10} = 0.125$, 27% $CrI$ inside ROPE).

Thus, we provide clear evidence that sounds modulated the first visual cortical response in congenital cataract reversal individuals but did not affect the first visual cortical response in any of the control groups. These results suggest bottom-up crossmodal activity in early visual cortex in humans with reversed congenital cataracts.

**Inverse modeling indicates $\Delta C1$ loci consistent with an early visual cortical origin.** Numerous previous source localization studies have reported the V1 to be the major contributor to the C1 wave[35,47,50]. To ascertain whether the neural generators of the $\Delta C1$ in the present data were consistent with an early visual cortical origin, the sLORETA[51] method was used to calculate source maps from the $\Delta C1$ topographies (see Methods: Source Modeling). Due to a sparse sampling of the scalp surface ($n_{electrodes} = 32$), the lack of individual MRI templates for forward modeling, and the relatively small amplitude of $\Delta C1$, we applied sLORETA on grand average $\Delta C1$ topographies to maximize signal-to-noise ratio as has been reported in studies with similar paradigms[35,47]. Except for the spatially congruent audiovisual condition ($AV_c$) in the CC group,

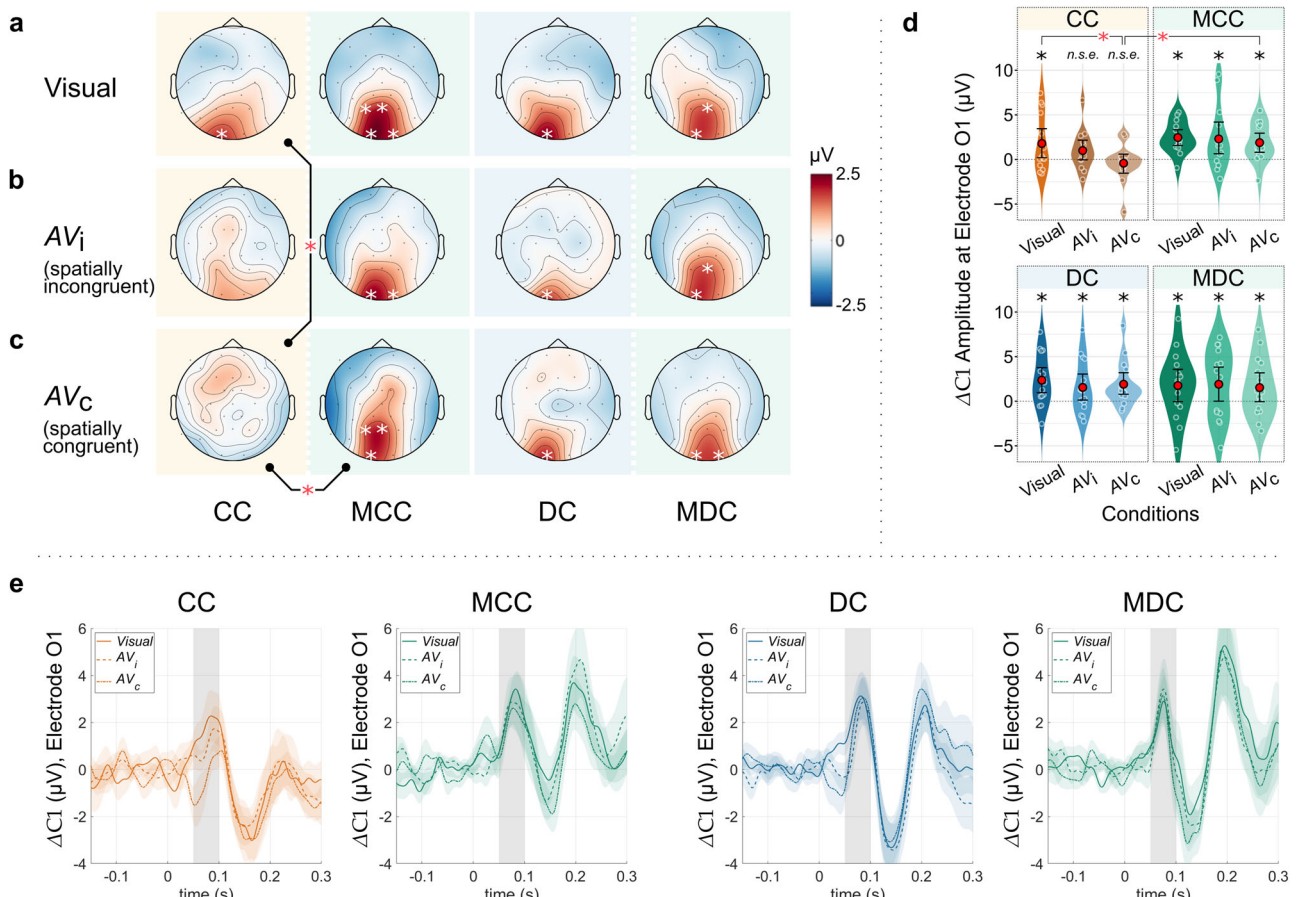

**Fig. 2 C1 difference wave (ΔC1) topographies and amplitudes at electrode O1 for the visual (V) and audiovisual (AV) stimulation conditions, presented separately for all four groups: congenital cataract reversal individuals (CC), matched controls for the CC group (MCC), developmental cataract reversal individuals (DC), and matched controls for the DC group (MDC). a** ΔC1 topographies for the unimodal visual (V) condition. Substantial evidence for a ΔC1 was found in all four groups (white stars: substantial evidence for a ΔC1, $BF_{+0}$ >3, and 95% credible interval (CrI) falling outside of the region of practical equivalence (ROPE)). **b** ΔC1 topographies for the spatially incongruent audiovisual condition ($AV_i$). **c** ΔC1 topographies for the spatially congruent audiovisual condition ($AV_c$). For both the $AV_i$ and the $AV_c$ conditions, substantial evidence for a ΔC1 was found in all three control groups (MCC, DC and MDC groups), but not in the CC group. The CC group's ΔC1 was additionally substantially diminished in the $AV_c$ condition compared to the MCC group, as well as compared to the V condition in the CC group. (Black bars, red stars: $BF_{10}$ >3, 95% CrI outside of ROPE). **d** ΔC1 amplitudes at electrode O1. Asterisks (*) indicate substantial or stronger evidence ($BF_{+0}/BF_{10}$ > 3) over the null hypothesis models, as well as positive ROPE tests, n.s.e. indicate no substantial evidence of a ΔC1. Red circles indicate the mean values, and the error bars represent 95% confidence intervals for the means, obtained by smoothed bootstrapping with Gaussian kernels. Individual data points have been jittered for readability ($n_{CC} = n_{MCC} = 14$, $n_{DC} = n_{MDC} = 15$, independent individuals). **e** Time course of the ΔC1 at electrode O1 for all tested groups (CC/MCC and DC/MDC), and stimulation conditions (V, $AV_i$ and $AV_c$). Error bands represent the standard error of the mean. Grey bars indicate the 50–100 ms range for ΔC1 employed in the analyses. Note that for easier visualization of a ΔC1 suppression, the waves are plotted with the y-axis positive upwards.

where the ΔC1 was suppressed, we observed source loci of the ΔC1 topographies consistent with an early visual cortical origin (Fig. 3a). Moreover, a left (contralateral) bias was observed for most of the sLORETA maps (Fig. 3a, b). In addition, the grand average sLORETA solutions were consistent with the ERP finding that the ΔC1 was modulated by spatial congruence in the CC group.

In a further exploratory step, we averaged across all ΔC1 source maps, excluding the $AV_c$ condition in the CC group ($n = 11$). In this second-level grand average source map, most of the vertices with the highest source values (>95% of the maximum amplitude) were located inside the left V1 (Fig. 3c), although the vertex with maximum activity was found in the early visual area V2 (Brodmann Area 18, Talairach coordinates: −12, −85, −17). The present inverse modeling results are consistent with the well accepted hypothesis that the ΔC1 has an contralateral early visual cortical origin, with a major contribution from V1[35,37,52].

**Unimpaired basic multisensory integration after sight restoration.** Audiovisual targets elicit faster responses compared to auditory or visual targets alone, a phenomenon known as the multisensory redundant target effect (Fig. 4a, b). A part of the observed faster reaction times, however, can be explained by simple statistical facilitation afforded by multiple information channels in the presence of noise. We tested violation of the race model inequality (RMI) with reaction times obtained from the target stimuli[39–41] to test whether sight-recovery individuals (the CC and the DC group) exhibited audiovisual integration beyond statistical facilitation (Methods: Behavioral Data Analysis), and if so, whether the amount of integration would be comparable to their matched control groups. The reaction time benefits conferred by audiovisual target stimuli where both the auditory as well as the visual stimulus were targets ($A^T V^T$), compared to audiovisual targets where only the auditory ($A^T V^0$), or only the visual stimulus ($A^0 V^T$) was a target, served to test the RMI

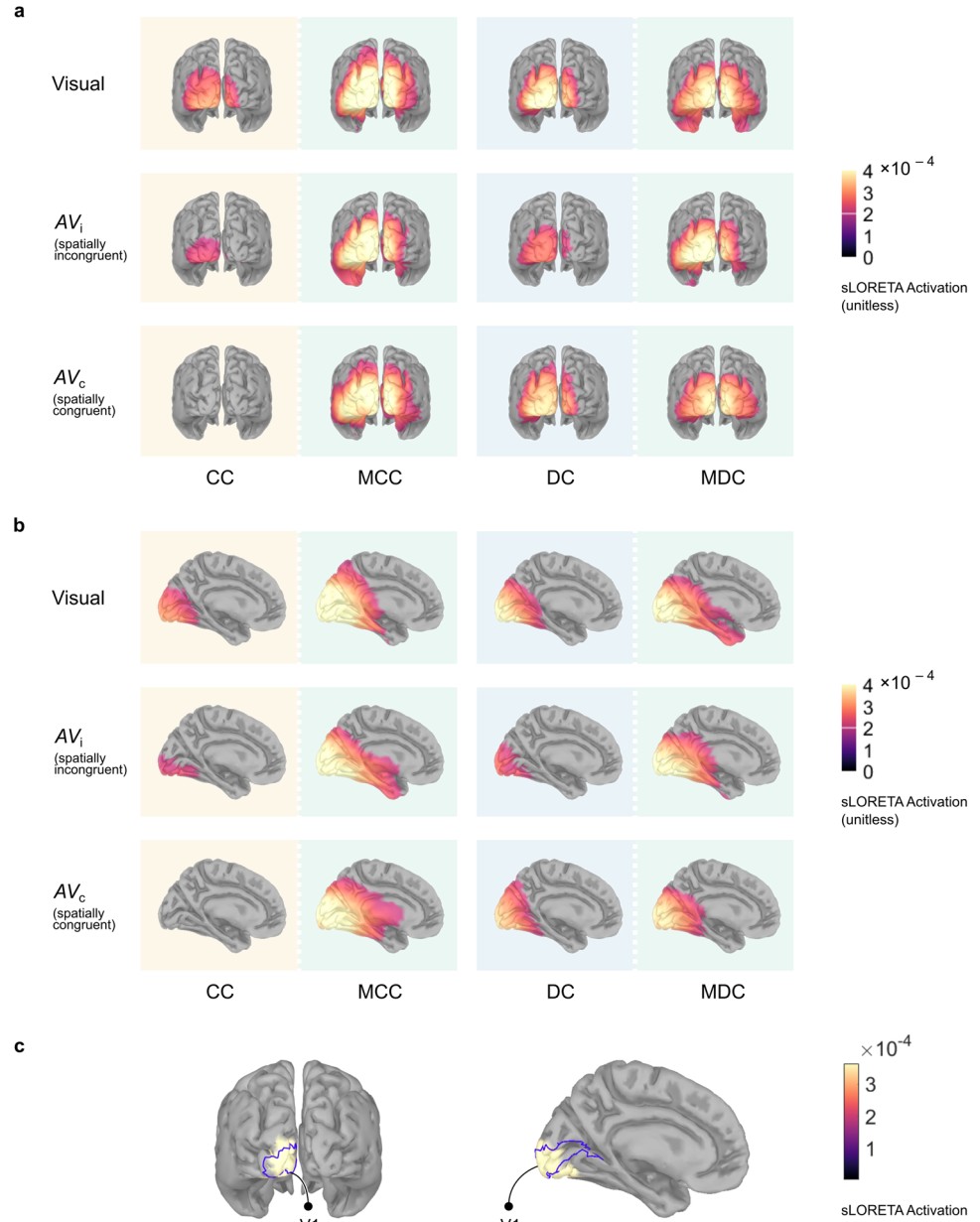

**Fig. 3 Standardized source current density (sLORETA solution) for the ΔC1 on the cortical surface for the visual ($V$), audiovisual incongruent ($AV_i$), and audiovisual congruent ($AV_c$) conditions, presented separately for all four groups: congenital cataract reversal individuals (CC), matched controls for the CC group (MCC), developmental cataract reversal individuals (DC), and matched controls for the DC group (MDC). a** sLORETA maps (posterior view) respectively for the grand average visual ($V$), spatially incongruent audiovisual ($AV_i$), and spatially congruent audiovisual ($AV_c$) conditions. **b** Identical maps (medial view), displayed on the cortical surface contralateral to the visual stimuli (left). Source maps in **a** and **b** were thresholded at 50% amplitude. **c** Mean of the sLORETA solutions for all ΔC1 conditions across all groups, except for the $AV_c$ condition in the CC group ($n = 11$). Source maps were thresholded at 95% of the maximum source current density. V1 location is shown overlaid.

inequality. To this end, first the nonnegative RMI violation area, a marker of audiovisual integration beyond chance level, was derived for spatially incongruent ($AV_i$) and spatially congruent ($AV_c$) target combinations separately in each participant group (see *Methods*). We next tested whether the RMI violation areas were distributed further away from zero than would be predicted by a prior exponential distribution with the same variance as the data (Fig. 4c). In all groups and spatial congruence conditions, we found strong evidence of race model violations (Bayesian linear mixed models, minimum $BF_{10,(CC/MCC)} = 20.97$, minimum $BF_{10,(DC/MDC)} = 62.67$, all of the 95% credible intervals fell outside of the region of practical equivalence, ROPE test). This evidence

was additionally ascertained with traditional non-parametric analysis methods[39] (see Supplementary Note 4, Supplementary Table 3, and Supplementary Fig. 5, all $ps < 0.05$, max. $p = 0.045$, cluster-based permutation tests, followed by Benjamini–Hochberg correction for multiple comparisons). Additionally, no substantial evidence was found that the RMI violation areas varied by groups and/or spatial congruence (Bayesian linear mixed models, $BF_{10}$ for intercept, CC/MCC: 633.35, DC/MDC: 26.39, positive ROPE tests; for both factors i.e., *Group* and *Spatial congruence*, and their interaction, $BF_{10} < 1$).

The behavioral results provide strong evidence that the CC group's reaction times benefited from audiovisual integration

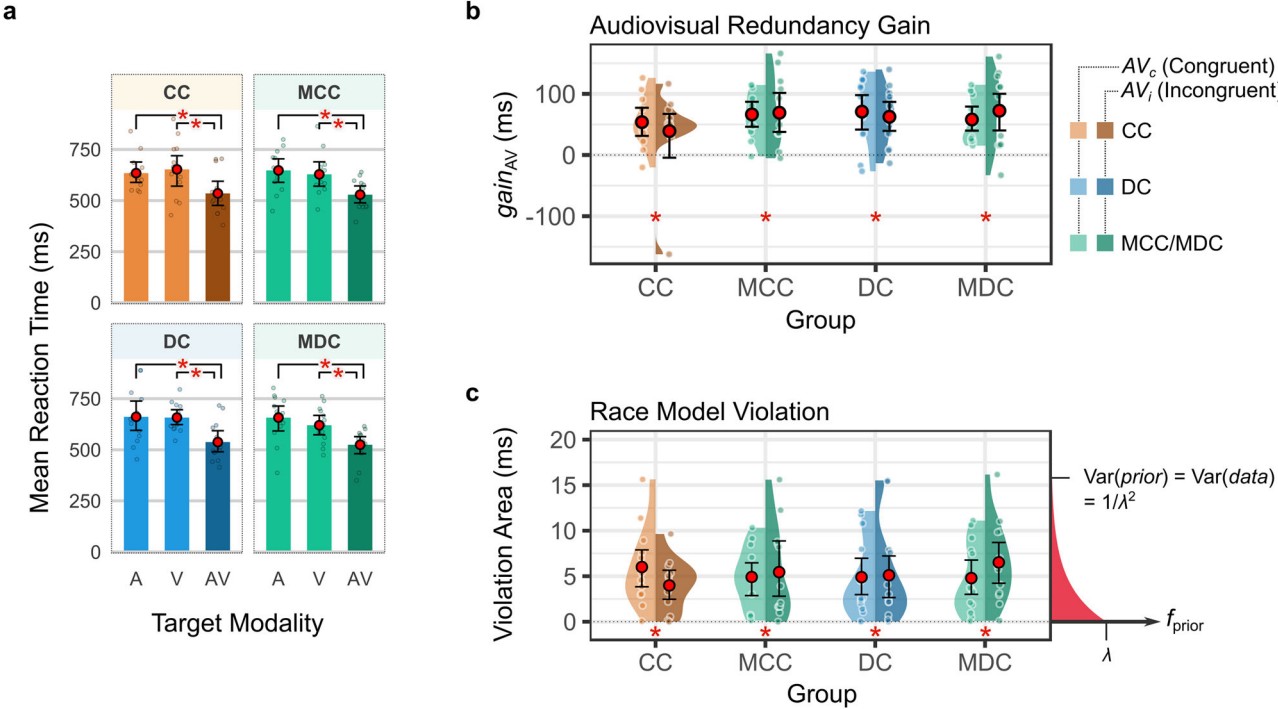

**Fig. 4 Analysis of reaction time improvements through audiovisual integration after sight recovery. a** Bar plots of reaction times to unimodal vs. bimodal targets. Target modalities, A: unimodal/bimodal stimuli where only the auditory stimulus was a target, V: unimodal/bimodal stimuli where only the visual stimulus was a target, AV: bimodal targets where both the auditory and the visual stimuli were targets. **b** Violin plots of audiovisual redundancy gain ($gain_{AV}$), defined as the gain (in ms) in mean reaction time to AV targets compared to the fastest of the A and the V targets (*Methods*). **c** Violin plots of nonnegative race model violation areas (in ms), a marker of multisensory integration beyond chance level. Race model violation areas were distributed away from zero compared to an exponential prior possessing the same variance as the data (prior for the CC/MCC analysis shown in red). In all subplots, asterisks (*) indicate substantial or stronger evidence ($BF_{+0}/BF_{10} > 3$) over the null hypothesis models, as well as positive ROPE tests. Red circles indicate the mean values, and the error bars represent 95% confidence intervals for the means, obtained by smoothed bootstrapping with Gaussian kernels. Individual data points have been jittered for readability ($n_{CC} = n_{MCC} = 12$, $n_{DC} = n_{MDC} = 13$, independent individuals).

akin to sighted controls. These findings replicate and extend previous results by showing that multisensory redundancy gains in target detection times recover even after extended periods of congenital visual deprivation[43,44].

## Discussion

Crossmodal activation of the visual cortex in permanently blind humans is a well-established finding[2,6–11,53,54]. Yet it has been unknown whether stimulus-driven auditory activity in early visual cortex retracts following sight restoration in congenital blind individuals. The present study employed event-related potentials in a paradigm that allowed us to target the earliest retinotopic visual cortical activity in isolation[35,46]. We demonstrate that when visual stimuli were accompanied by concurrent auditory stimulation, their processing was suppressed in sight-recovery individuals who were born pattern vision blind (CC group). In stark contrast, concomitant auditory stimulation did not affect bottom-up early visual processing in typically sighted controls, nor in a group of sight-recovery individuals who had suffered from a period of transient developmental (late onset) visual impairment (DC group) instead of congenital blindness.

Retinotopy, the existence of well-defined topographic representations of the visual field, is a hallmark of visual cortical organization, and the $\Delta C1$ is the earliest known electrophysiological marker of retinotopic visual organization and bottom-up visual function in V1[34,35,52]. The $\Delta C1$ reduction in the CC group was substantial for spatially congruent audiovisual stimulation compared to unimodal visual stimulation. The results

provide strong evidence that crossmodal activity of the visual cortex does not completely retract after sight restoration following congenital blindness, resulting in auditory information modulating the earliest, retinotopic visual cortical processing in the CC group. In contrast, concurrent auditory stimulation did not affect the $\Delta C1$ in the typically sighted control groups (MCC/MDC), nor in the DC group.

Potential explanations for an auditory influence on the bottom-up activity at the earliest visual cortical processing stage in the CC group require consideration of three interrelated anatomical and physiological aspects. First, such explanations must respect temporal constraints for auditory information flow to the visual cortex, and second, need to provide parsimonious structural accounts for the information flow. Last and equally important, they must consider the effects of a period of transient congenital blindness on creating new structures and pathways, or modifying existing ones, for auditory information flow to V1.

A median latency of about 20 ms (rounded up to the nearest multiple of 5 ms) has been reported for broadband auditory stimuli to reach the primary auditory cortex (A1) in macaques[55], whereas flash stimuli arrive at the macaque's V1 with a median latency of about 65 ms[56]. These data agree well with direct electrocorticographic recordings in the human A1 (median: ~10–25 ms) and V1 (onset: ~55 ms, peak: ~95 ms)[57,58], as well as with the $\Delta C1$ as a noninvasive marker of the earliest stimulus driven visual cortical processing (onset around 50 ms[35,47]). For sounds to influence visual processing in the V1, a window of around 45 ms (=65 ms for visual − 20 ms for auditory cortical arrival times) thus appears reasonable. Based on this temporal

consideration, we exclude feedback from higher multisensory areas as the main driver of the $\Delta C1$ suppression. For example, the multisensory lateral intraparietal area (LIP) exhibits a median latency of ~100 ms to sounds, which would be too late[59] for affecting the C1 wave elicited by a concurrent visual stimulation.

Direct thalamocortical connections to the V1, either from the first-order auditory, or (possibly crossmodally rewired) visual thalamic nucleus provides a hypothetical alternative for auditory information flow to V1. A comprehensive recent work in a Mongolian gerbil model has reported an exuberance of thalamocortical connections to the V1 from visual as well as nonvisual thalamic nuclei early in life[60], which are substantially pruned during typical development. Bilateral enucleation before eye opening was reported to lead to a strong increase of visual as well as nonvisual thalamocortical connections in the gerbils' V1, including from higher-order auditory and multisensory thalamic nuclei. Critically, no evidence for a direct input from the medial geniculate nucleus (MGN, the first-order auditory thalamic nucleus) to V1 was found in the blind Mongolian gerbils[60]. These results mesh well with those in enucleated mice which did not find direct MGN→V1 projections[61], and a report of scarce (a few in 1 out of 8 animals) MGN→V1 projections in opossums enucleated before the visual pathway was firmly established[19]. Dynamic causal modeling in permanently congenitally blind humans, as well as in sight-recovery humans in fMRI studies have additionally found no evidence for a stronger MGN→V1 connection compared to normally sighted participants[22,28]. Taken together, neither the results in human studies, nor those from non-human animal models support the idea of direct thalamocortical routes from the MGN to the V1 to explain the $\Delta C1$ suppression as observed in the present study.

An alternative possibility is that a crossmodally rewired lateral geniculate nucleus (LGN), the first-order visual thalamic nucleus, might receive auditory input and relay this information to the V1, as has been reported in blind mole rats[62], congenitally anophthalmic mice[63,64], and in hamsters enucleated at birth[65]. A recent magnetoencephalographic (MEG) source modeling study involving permanently congenitally blind humans has reported a very early occipital activity evoked by tactile stimulation (onset ~35 ms)[11], likewise arguing that a direct connection between the first-order somatosensory nucleus and the LGN might similarly provide a fast path for crossmodal activation in the occipital cortex of congenitally blind humans.

A common feature of the non-human animal studies reporting crossmodal rewiring of the LGN is the disruption of the maturation of the pathway from the retina to the LGN: in blind mole rats, the optic nerves undergo programmed postnatal degeneration[62,66], in congenitally anophthalmic mice strains, the optic nerves do not develop[61,63], and unlike in newborn humans, the retinal ganglion cells of newborn hamsters have not established projections but just started growing their collaterals into the LGN[67]. As the early patterning of the thalamus is intricately choreographed by chemical gradients and cell signaling[68–70], it has been argued that these patterns of crossmodal plasticity, obtained from specific animal models with disrupted chemical gradients, might not be the norm in most cases[61]. Accordingly, in a study involving neonatally enucleated mice (as opposed to the congenitally anophthalmic mice in the same study), no auditory innervation of the LGN was demonstrated. The authors suggested that prenatal spontaneous retinal activity might be sufficient to permanently visually imprint the LGN[61]. In congenitally blind humans, an fMRI study which demonstrated considerable crossmodal activation of the visual cortex likewise did not report any crossmodal activation of the LGN[71]. In precocious animals like humans, where the (subcortical) visual system is comparably more developed at birth[46,72,73], a direct connection to the LGN

from another first-order thalamic nucleus thus seems unlikely[68]. In a rat model, direct connections between the thalamic nuclei have moreover been reported to be at most very sparse[74]. Likewise for humans, it has been argued that thalamo-cortico-thalamic loops and the reticular nucleus, which controls the transmission of thalamic activity to the cortex[75], offer likelier communication pathways between the thalamic nuclei[68,74,76].

Instead of direct connections between the first-order thalamic nuclei, exuberant projections to V1 from higher order auditory/multisensory thalamic nuclei, as suggested by Henschke et al.[60] in Mongolian gerbils enucleated shortly before eye opening, might provide an alternative pathway involving the thalamus (cortico-thalamo-cortical loops). The pulvinar, the largest thalamic nucleus, might be considered a candidate for possible thalamocortical routes to V1 since it fulfills all temporal and structural constraints. The pulvinar is a higher-order thalamic nucleus which receives and processes input from multiple sensory modalities, and is known to project to early visual cortex including V1[77]. Moreover, the pulvinar has been recently reported to strengthen its projections to V1 after in utero enucleation in macaques[78]. The pulvinar's ability to selectively gate, i.e., enhance or suppress early visual cortical activity, as reported in nonhuman primates, makes it a promising node which might influence the $\Delta C1$ suppression in the CC group[79].

A second candidate for auditory information flow to V1 would be direct cortico-cortical connections, e.g., between the A1 and the V1, as has been reported in rodents and primates alike[23,24]. These connections exist in typically sighted animals as well, and dynamic causal modeling of fMRI connectivity has suggested a strengthened cortico-cortical connection between the A1 and the V1 due to congenital visual deprivation[22,80]. The methods employed in the present study do not allow a conclusive decision about which exuberant connectivity, i.e., a cortico-cortical or cortico-thalamo-cortical (or both) contribute to the suppression of the $\Delta C1$ by auditory stimuli. Although recent evidence has indicated that subcortical electrophysiological activity might be detectable with scalp EEG[81], a considerably higher electrode density as well as number of trials, combined with individual MRI templates would be likely necessary to evaluate the alternative models in sight-recovery humans. These steps would pose a major challenge in rare clinical samples such as in our study.

We would like to underscore that cortico-cortical and cortico-thalamo-cortical explanations for the crossmodal activation of the V1 do not need to be mutually exclusive. Recent views on the pulvinar consider this nucleus to be involved in coordinating and synchronizing large cortical networks, e.g., during attention employment and multisensory processing[82,83]. Deactivation of the pulvinar, for example, has been reported to abolish V1 activation, and selective activation of the pulvinar can enhance V1 response in specific areas while suppressing adjacent regions[79]. Moreover, it has been suggested that for a direct connection between two cortical areas, generally there is another indirect path passing through the pulvinar (the replication principle)[77,84]. Reports that higher speech perception abilities in blind individuals was accompanied by activations of not only the A1, but the V1 and the pulvinar as well, further corroborate this account[85].

An exploratory analysis in the current study indicated that the $\Delta C1$ was suppressed in a spatially specific manner by concurrent auditory stimuli in the CC group. While the central auditory system receives information from both ears already at the level of the brainstem[86], a contralateral bias for auditory stimuli has been reported for multiple auditory brain structures including the MGN and the auditory cortex[87,88]. In the present study, we observed no auditory-information-driven $\Delta C1$ suppression in the control groups, yet a spatially specific effect of auditory information on the $\Delta C1$ in the CC group. In the CC group, the spatial

specificity additionally indicates that the observed $\Delta C1$ suppression is likely not caused by a general, unspecific suppression of early visual cortical activity by auditory processing and might indicate that the typical exuberant crossmodal connectivity features some spatial (hemispheric) selectivity.

Despite sound-driven suppression of the first visual cortical activity, in our study the CC group showed robust multisensory benefits for reaction times with redundant audiovisual stimuli, replicating two previous reports of the multisensory redundant target effect being spared in independent groups of sight-recovery individuals[43,44]. The multisensory improvements exceeded chance-level facilitation and were indistinguishable from their sighted control groups, suggesting that bottom-up integration of auditory and visual input in V1 might not be a prerequisite for faster responses to crossmodal stimuli. This hypothesis is compatible with reports suggesting that the crossmodal redundant target effect arises in subcortical areas such as the superior colliculus, which has been indicated to drive the learning of basic multisensory integration in dark-reared cats[89]. Subcortical mechanisms have been similarly hypothesized to underlie auditory stimulation driven visual behavioral improvements in hemianopic patients with visual cortical lesions[90], but has heretofore never been unambiguously demonstrated in healthy individuals. In the typically sighted MCC and the MDC groups, as well as in the DC group, a modulation of retinotopic bottom-up activity in early visual cortex was not observed despite these groups exhibiting crossmodal multisensory gain, further supporting the hypothesis that multisensory reaction time benefits do not depend on an early modulation of retinotopic V1 activity by sounds.

In a combined analysis of the typically sighted control groups (MCC and MDC groups), concurrent auditory stimulation did not modulate the $\Delta C1$ regardless of the spatial congruence of the auditory and visual stimuli (Supplementary Note 3). This lack of $\Delta C1$ modulation was observed despite robust multisensory reaction time gains conferred by auditory stimuli. To the best of our knowledge, the present study provides the first report that concurrent and task-relevant auditory information, which led to substantial multisensory benefit beyond chance level, does not modulate the earliest retinotopic visual cortical activity in typically sighted controls. Whether the first feedforward retinotopic activity in V1 (as indexed by the $\Delta C1$) is modulated by attention is still an ongoing debate after fifty years of the discovery of the C1 wave[34,52], with most studies indicating that an attentional modulation of the $\Delta C1$, if present at all, is likely small[91]. Based on the results in the typically sighted control groups of the present study, we add the hypothesis that the first sweep of feedforward retinotopic visual cortical activity might be unaffected by a range of concurrent nonvisual sensory processing. This hypothesis is compatible with recent works in perceptual decision-making contexts which found evidence for a later audiovisual integration, after the initial feedforward stages[92,93], but stands in contrast to a report of early (<100 ms) audiovisual integration in the visual cortex recorded with magnetoencephalographic (MEG) techniques[94]. The lack of $\Delta C1$ modulation by concurrent auditory processing is a critically important finding of a possible constraint in the context of multisensory integration of redundant stimuli in typically sighted humans, because V1 is reported to contain numerous multisensory neurons and receives direct projections from the auditory cortex[24,95,96].

In the DC group, the existence of a $\Delta C1$ across all conditions, and the observation that they were not substantially different from their controls, the MDC group, exclude specific testing environment differences or participant ethnicity as possible explanations of the CC vs. MCC group differences. Moreover, they provide evidence that the observed effects in the CC group cannot be attributed to unspecific effects of suffering from or being treated for cataracts, nor to visual impairments emerging at some time during development.

Finally, the existence of a $\Delta C1$ for the unimodal visual condition in the CC group replicates our previous results from an independent study[46] and underscores the robustness of lower-level visual processing to often extended periods of pattern vision deprivation. At the same time, $\Delta C1$'s suppression in audiovisual conditions requires considering visual processing after sight recovery from the standpoint of living in a multisensory world, especially the possible effects of noise and conflicting auditory information on vision after sight restoration[33,97]. Crossmodal plasticity after sight recovery can modulate visual perception[33], and could lead to reduced performance in higher-level multisensory integration and speech perception[15,42]. The combined evidence indicates that for optimal rehabilitation, the altered neural landscape of sight-recovery individuals must be taken into consideration[16,98].

In conclusion, to the best of our knowledge we provide the first report of persistent crossmodal bottom-up modulation of the earliest visual cortical response in a rare group of humans who were born pattern vision blind, and subsequently gained vision. We suggest that atypical crossmodal brain networks, likely acquired in a sensitive period early in life during a phase of congenital blindness, are not completely lost after sight recovery, but coexist with spared visual networks, and can modulate very early visual processing.

## Methods

**Participants**. Thirty-one sight-recovery individuals with a history of visual deprivation through bilateral cataracts took part in the study. Fourteen of them had suffered from dense complete bilateral congenital cataracts before undergoing cataract-reversal surgery (CC group; mean age at surgery/mean duration of pattern vision blindness: 42.14 months, range = 1 month – 17.75 years). Ten of the 14 CC individuals, who were operated at the L V Prasad Eye Institute and for whom pre-surgical acuity measurements were available, were all blind at the time of presentation (Category 5, ICD11 – 9D90.4, Supplementary Note 5, and Supplementary Table 4)[99]. The mean age of the 14 CC participants was 17.07 years (range = 6–39 years). One participant was left-handed, and 3 were female. The mean time since surgery at testing was 13.92 years (range = 3.33–37 years). The CC group had a geometric mean visual acuity of 0.229 (decimal, range = 0.051–0.7, see Supplementary Table 4 for details).

The 15 other analyzed participants had bilateral, though not necessarily complete, developmental cataracts (DC group, see Supplementary Table 5 for details) and underwent the same surgical procedure to restore vision. They were on average 14.47 years old (range = 9–24 years) and were operated at the mean age of 7.36 years (range = 1.92–14.42 years). The mean time since surgery was 7.52 years (range = 4.58–22 years). Four DC participants were female, and one was left-handed. The DC group had a geometric mean visual acuity of 0.619 (decimal, range = 0.23–1.00, see Supplementary Table 5 for details), with decisive evidence for higher visual acuity compared to the CC group (Bayesian one-sided independent samples $t$-test after converting the acuities to LogMAR values, $BF_{+0} = 483.17$; Cohen's $d = 1.769$, 95% CI = [0.869, 2.668]). We found no substantial evidence for an age difference between the CC and the DC groups ($BF_{10} = 0.229$). We excluded two participants from analysis: one due to a lack of etiological certainty (i.e., congenital vs. developmental origins), and the other because of a history of neurological and developmental disorders.

Participants were included only when a very high degree of confidence in their diagnosis was indicated by a panel involving ophthalmologists and optometrists based on medical records and clinical examinations. The initial screening procedure included participants who had bilateral dense cataracts rendering the fundus invisible at the time of presentation, or if partially absorbed hypermature cataracts accompanied with sensory nystagmus were confirmed. Behavioral/family history data from the patients and their immediate family members, caregivers and/ or healthcare providers were additionally collected and triangulated. The existence of sensory strabismus (e.g., esotropia) was used as an additional classification criterion in combination with the other information[100]. Recently, we have reported that electrophysiological signatures of extrastriate visual processing, which did not constitute a classification criterion for the present sample, could cluster the CC individuals in the present study in line with expert panel diagnosis[16].

For each sight-recovery individual we tested a control participant matched for age, sex, and handedness ($n = 29$, mean age = 15.66 years, range = 7–38 years, 7 female, 2 left-handed). All of them had normal or corrected-to-normal vision and had no history of sensory problems.

Adult participants received a small monetary compensation for taking part in the study, and minors received a small present. Both the congenital and developmental cataract reversal individuals were tested at the L V Prasad Eye Institute (LVPEI), Hyderabad, India; control participants were recruited from the local community of Hamburg, Germany. The study was jointly approved by the local ethical commission of the LVPEI and of the faculty of Psychology and Human Movement Science at the University of Hamburg and conformed to the ethical principles of the Declaration of Helsinki (2013). Written informed consent was obtained from all participants. A legal guardian additionally provided written informed consent for minors. All participants were healthy, except cataract-related visual impairments, and did not have any neurological problems according to self-reports, guardian assessments, or in the case of cataract reversal individuals, an additional general clinical assessment.

**Stimuli.** Visual stimuli were circles, 2.5° in diameter, containing full-contrast square wave gratings with a spatial frequency of 2 cycles/degree. Horizontal patterns ($P = 80\%$) served as standard stimuli, whereas vertical patterns ($P = 20\%$) served as rare deviant targets. The stimuli were presented for 150 ms, one at a time in one of the four visual field quadrants, at an eccentricity of 4°, and an angle of 25° for upper visual field locations (UVF) and of −45° for lower visual field locations (LVF). A Dell IN2030 monitor was used at the LVPEI and a Samsung P2370 monitor was used at the University of Hamburg. Both operated at a refresh rate of 60 Hz (nominal luminance: 250 cd/m²).

Auditory stimuli were 65 dB (A-weighted), 150-ms-long approximate white noise bursts. Standards ($P = 80\%$) were continuous noise bursts, whereas deviants, serving as behavioral targets, were noise bursts containing a 16.67 ms period of white noise followed by a 16.67 ms period of silence, repeating to a total duration of 150 ms ($P = 20\%$). Auditory stimuli came from a loudspeaker on either the left or the right side of the participant, placed directly below the screen (See Fig. 1c). Stimulus duration, audiovisual synchrony, and triggering latency were ensured with an in-house measurement solution with a photodiode (SFH-203, Osram Opto Semiconductors GmbH, Regensburg, Germany) and a microphone (BOB-12758, SparkFun Electronics, CO, USA).

The stimuli were presented with the PsychoPy framework (v1.83)[101].

**EEG data acquisition.** EEG data were continuously recorded from 32 electrodes fixed in a custom EASYCAP recording cap (electrode locations were, in standard 10/20 system: FP1, FP2, F7, F3, Fz, F4, F8, FC5, FC1, FCz, FC2, FC6, T7, C3, Cz, C4, T8, TP9, CP5, CP1, CP2, CP6, TP10, P7, P3, Pz, P4, P8, O1, O2, F9, and F10). Passive Ag/AgCl electrodes were used with the left earlobe serving as the online reference. Recording was performed through BrainVision BrainAmp DC/MR Amplifiers (Brain Products GmbH, Gilching, Germany), with the following settings: lower cutoff frequency = 0.016 Hz, upper cutoff frequency = 250 Hz, sampling rate = 1 kHz.

**EEG preprocessing.** Preprocessing was performed with EEGLAB version 11.5.4b[102], running on MATLAB version 2012b (Math-Works, Natick, MA, USA), employing in-house MATLAB scripts. We average-referenced the data offline and notch-filtered electrical line noise artifacts, if present, at 50 Hz and its multiples. Biological artefacts e.g. blinks, saccades, ECG, and conspicuous muscle activities were marked and removed with independent components analysis (ICA). To ensure that event-related potentials (ERPs) would not be confounded by eye movements or blinks at the time of stimulus presentation, we rejected stimulus epochs with blinks or eye movements in the time window of −25 ms to 175 ms with respect to the stimulus presentation. To this end, we separated the ICA components corresponding to blink and eye movements to create a separate data file. Ocular artifact thresholds were defined in these data as 5 standard deviations of the maximal activities of the frontopolar electrodes FP1 and FP2 for detecting blinks, and 3 standard deviations of the maximal activities of the electrodes F9 and F10 for detecting saccades/eye movements[16]. This conservative procedure ensured that C1 difference waves would not be artificially lowered by blinks or eye movements during stimulus presentation. Additionally, epochs with a button press by participants within 500 ms of a stimulus presentation were rejected to avoid contamination from motor artefacts. Subsequently the EEG data were bandpass filtered with a lower cutoff frequency of 0.1 Hz and a higher cutoff frequency of 40 Hz.

For stimuli that contained a left visual field stimulation, we mirrored the data across the nasion-inion axis, by swapping the electrodes' recordings with the corresponding ones in the mirrored location, leaving the midline electrodes untouched, and collapsed them with the corresponding stimuli containing a right visual field stimulation (see also Supplementary Note 1). This step effectively provides twice the number of stimuli per condition[16,46]. For unimodal auditory stimulation, we similarly mirrored the auditory left conditions and collapsed them with the auditory right conditions. The resulting epochs were baselined with a window of −100 ms to 0 ms, and ERPs were then derived for each of the collapsed six conditions containing a visual stimulus : visual ($V$; $V_{UVF}$, $V_{LVF}$), audiovisual congruent, i.e. where auditory and visual stimulation both were from the left or the right side ($AV_c$: $AV_{c,UVF}$, $AV_{c,LVF}$), and audiovisual incongruent, i.e. where auditory and visual stimulation came from opposite sides ($AV_i$: $AV_{i,UVF}$, $AV_{i,LVF}$; for the individual C1 waves from the UVF and the LVF stimulation conditions, see Supplementary Note 6 and Supplementary Fig. 6). The $\Delta C1$ was calculated for the $V$, $AV_i$, and $AV_c$ categories by subtracting the upper visual field ERPs from the lower visual field ERPs, that is, $\Delta C1 \stackrel{\text{def}}{=} v_{LVF} - v_{UVF}$. The auditory ERPs are not reported in this study. Parts of the data from the unimodal visual condition have been included in a previous publication as a validation dataset for a biomarker ($V_{UVF}$, latency range: 120–170 ms)[16].

**ERP analysis**. The mean of the $\Delta C1$, calculated at each electrode in the time window of 50–100 ms after stimulus onset, was the dependent variable. For each sight-recovery group and their controls (e.g. CC/MCC and DC/MDC), we ran two Bayesian hierarchical models using the five posterior-occipital electrodes where the $\Delta C1$ is most prominent: O1, O2, P3, P4, and Pz. The first model was used for determining the existence of a $\Delta C1$, and the second to test for a-priori condition/group differences. Based on a pilot study (Supplementary Note 2), we ran custom contrasts instead of omnibus tests[49,103], as we expected no difference between the conditions in the MCC/MDC groups, and as many of the possible pairwise comparisons would be hardly meaningful (e.g. $\Delta C1_{MCC,AVc} - \Delta C1_{CC,V}$). In the first parameterization, we simultaneously tested five contrasts at each of the selected posterior-occipital electrodes. The first two of these contrasts tested whether the $\Delta C1$ in the audiovisual conditions differed from the $\Delta C1$ in unimodal visual stimulation in the CC group:

$$\Delta C1_{CC,V} - \Delta C1_{CC,AV_i} \text{ and } \Delta C1_{CC,V} - \Delta C1_{CC,AV_c}.$$

The next three contrasts investigated whether the $\Delta C1$ in each condition differed between the CC and the MCC group. These three contrasts were:

$$\Delta C1_{MCC,V} - \Delta C1_{CC,V}, \Delta C1_{MCC,AV_i}$$
$$-\Delta C1_{CC,AV_i} \text{ and } \Delta C1_{MCC,AV_c} - \Delta C1_{CC,AV_c}.$$

Necessary orthogonal contrasts were added to make the model full-rank[49]. An equivalent model was run for the DC/MDC group comparisons.

In the second parameterization, we estimated the $\Delta C1$ values for all separate conditions simultaneously at each electrode using a cell-means model. The presence of a $\Delta C1$ was ascertained using a one-sided Bayes factor indicating substantial evidence ($BF_{+0} > 3$) and when the 95% highest-density credible interval for $\Delta C1$ (95% CrI) fell outside a null region of practical equivalence, defined as 0.1 times the standard deviation of the data (ROPE test)[104]. We used one-sided BFs based on a strong expectation of the group mean of the $\Delta C1$ being nonnegative; to date, there has been no report of the $\Delta C1$ as formulated in this study being negative[16,34,35].

**Source modeling**. We used the sLORETA method for EEG inverse modeling[51] with the Brainstorm software package to derive source maps from voltage topographies[105]. The forward model was calculated using the DUNEuro partial differential equation solver package[106], with the New York Head model[107]. The inverse models were calculated with unconstrained source directions on a cortical sheet with 15002 voxels, and an identity matrix was used as the noise covariance. Due to a limited number of electrodes ($n_{elec} = 32$), the lack of individual forward models using MRI templates, and the $\Delta C1$ being a relatively small ERP wave, we applied sLORETA on the grand average $\Delta C1$ topographies to maximize the signal-to-noise ratio, as has been reported in studies with similar paradigms[35,47].

**Behavioral data analysis**. Participants responded to rarely presented targets ($P = 20\%$), which could be auditory, visual, or audiovisual. The task was unspeeded, and participants were instructed to prefer correct responses to reaction speed (Supplementary Note 4).

First, we investigated whether all participant groups reacted faster to bimodal targets with both auditory and visual target stimuli (AV targets $= \{A^T V^T\}$), compared to unimodal/bimodal targets where either only the auditory, or only the visual stimulus was a target. (Auditory targets $= \{A^T, A^T V^0\}$; visual targets $= \{V^T, A^0 V^T\}$, where $T$ denotes a target, and $0$, a standard stimulus). To this end, Bayesian linear mixed models were used, with AV targets as the reference group. Separate models were used for CC/MCC and DC/MDC comparisons.

*Redundancy gain*. Audiovisual redundancy gain was defined as

$$gain_{AV} = \min\left(\overline{RT(A^T)}, \overline{RT(A^T V^0)}, \overline{RT(V^T)}, \overline{RT(A^0 V^T)}\right) - \overline{RT(A^T V^T)}$$

that is, the speedup of mean reaction times to double bimodal targets compared to the minimum mean reaction time of the other target classes (unimodal auditory targets, unimodal visual targets, bimodal stimuli with only auditory targets, and bimodal stimuli with only visual targets). Bayesian linear mixed models with weak priors suggested by the *auto_prior()* function[108] were used to investigate the presence of audiovisual redundancy gain in each group, and to ascertain whether it differed between groups and by stimulus congruence. We used one-sided Bayes factors ($BF_{+0} > 3$), and additionally performed ROPE tests to confirm the presence of an audiovisual redundancy gain, because a slowing of reaction times due to bimodal double targets would be implausible based on extant reports in similar groups[43]. Separate models were used for the CC/MCC and the DC/MDC analyses.

*Race model violation area*. In the bimodal redundant target effect, participants react faster to targets of two modalities presented together compared to stimuli with a target of only one modality. The race model inequality (RMI) sets an upper bound for statistical facilitation obtained from two sensory channels[38,41] (see Supplementary Note 4 for details). This upper bound stipulates that for purely statistical facilitation of reaction times, the cumulative distribution functions of reactions times to auditory ($F_A$), visual ($F_V$), and audiovisual stimuli ($F_{AV}$) must obey the following inequality:

$$F_{AV}(t) \leq F_A(t) + F_V(t), \text{ where t is the reaction time.}$$

We took the nonnegative area between $F_{AV}(t)$ and $F_A(t) + F_V(t)$, that is, the integral of $\max[F_{AV}(t) - F_A(t) - F_V(t), 0]$, as a geometric measure of RMI violation[109,110]. Trivially testing these nonnegative values against the value of zero would be ill-motivated. We tested whether the distributions of the nonnegative RMI violation areas were sufficiently distributed away from zero than would be predicted by an exponential distribution possessing the same variance as the data. Moreover, we tested whether the nonnegative RMI violation depended on spatial congruence, or group, with Bayesian linear mixed models. As in all other analyses, separate models were employed for CC/MCC and DC/MDC group comparisons.

**Statistics and reproducibility**. The Bayesian hierarchical models used in ERP analysis were fitted with the *brms* package in Rv4.2.2[111,112]. In all cases, non-flat, weakly informative normal priors were used, whose standard deviations were set based on the *auto_prior()* function[108], taking 2.5 times the standard deviation of the data for non-intercept coefficients, and 10 times the standard deviation for the global intercept. Each Bayesian model drew 40,000 sample using a Hamiltonian Monte-Carlo sampler. Derived from proper (integrable to 1), non-flat priors and the use of Bayesian hierarchical (multilevel) models, the substantiality of the estimated parameters do not need correction for multiple testing[113,114]. To ensure reproducibility, the random number generators were seeded with a fixed seed prior to sampling commencement.

Based on the work of Gondan et al.[39,40], a non-parametric cluster-based permutation test was used to investigate RMI violations in the response time data (see also Supplementary Note 4, Supplementary Table 3, and Supplementary Fig. 5). In the permutation test, 10,001 draws between the 5th and 30th percentiles were used, in steps of 5 percentiles. The permutation test had a significance level, $\alpha = 0.05$ at the cluster level. In absence of a-priori hypotheses, the p-values of the cluster-based permutation tests were corrected with a Benjamini–Hochberg procedure in Rv4.2.2[112]. The false discovery rate for the procedure was $q = 0.05$.

**Reporting summary**. Further information on research design is available in the Nature Portfolio Reporting Summary linked to this article.

## Data availability

Aggregated, pseudonymized data have been deposited at the University of Hamburg research data repository (https://doi.org/10.25592/uhhfdm.13468)[115]. These data will be made available to external investigators upon reasonable request to the corresponding author through data transfer agreements approved by the stakeholders, under stipulations of applicable law including but not limited to the General Data Protection Regulation (GDPR; EU 2016/679). The source data behind the graphs in Figs. 2d, 4a–c can be found in Supplementary Data 1, 2, 3, 4 respectively.

## Code availability

Software code to replicate the results has been deposited at the University of Hamburg research data repository (https://doi.org/10.25592/uhhfdm.13468), with identical access criteria[115]. Here we additionally report the versions of software used to collect/process the data. Experiment programming: PsychoPy v1.83[101]. Data recording: BrainVision Recorder v.1.20 (BrainVision LLC, Garner, NC, USA). EEG preprocessing: MATLAB v.2012b (MathWorks, Natick, MA), EEGLAB v.11.5.4b[102]. R analysis: R (v.4.2.2)[112], attached packages(_versions): effsize_0.8.1[116], MASS_7.3–58.3[117], bayesplot_1.10.0[118], kernelboot_0.1.9[119], bayestestR_0.13.0[120], sjmisc_2.8.9[121], sjstats_0.18.2[108], brms_2.19.0[111], Rcpp_1.0.10[122], readxl_1.4.2[123], nlme_3.1–162[124], R.matlab_3.7.0[125], ggplot2_3.4.2[126], tidyr_1.3.0[126], dplyr_1.1.2[126]. Source analysis: sLORETA implemented in Brainstorm v.15-Aug-2023[105], running on MATLAB v.R2022a (MathWorks, Natick, MA, USA).

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

## Acknowledgements

We are grateful to D. Balasubramanian of the L V Prasad Eye Institute for supporting our study, and thank Seema Banerjee, Larissa Brockmann, Maria Guerreiro, Marlene Hense, Giulia Dormal, Siddhart Srivatsav Rajendran, Lisa Stockleben, and Florian Süßer for helping with data acquisition. Additionally, we thank Matthias Gondan-Rochon for a discussion related to the analysis of reaction times. The study was funded by the European Research Council grant ERC-2009-AdG 249425-*CriticalBrainChanges* and DFG Ro 2625/10-1 to B.R.

## Author contributions

S.S., D.B. and B.R. designed the experiments and planned the study. S.S. programmed the experiment and analyzed the data. S.S., R.K., D.B., I.S., K.P., and B.R. were involved in the recruitment and classification of participants, and in writing and revising the manuscript. S.S., R.K., D.B., I.S., K.P., and B.R. reviewed and approved the final version of the manuscript.

## Funding

## Competing interests

The authors declare no competing interests.
