## [Peer Review File · Communications Biology]

Reviewers' comments:

Reviewer #1 (Remarks to the Author):

This study uses visual ERPs to investigate whether and how the well-documented effect of cross-modal (auditory) activation in early visual cortex (V1) of congenitally blind humans (bilateral congenital cataract, CC group) changes as a function of sight-recovery after surgery. The effect is compared to a group of participants with surgically reversed developmental cataract (DC group), as well as normal-sighted control groups, matching the cataract-groups (MCC, MDC). To assess cross-modal activation in early visual cortex, ERPs elicited by gratings presented at a fixed position in one of the four visual quadrants are analyzed. The positions are chosen to optimally derive the C1 component - a modulation of the VEP known to reflect activity generated in primary visual cortex. Cross-modal activation in V1 is assessed by co-presenting sounds with the gratings either from a congruent or incongruent spatial direction (sound appearing at the side of, or opposite to the grating). It is reported that unimodal visual stimulation elicits a clear C1 response in all groups. Cross-modal stimulation elicits a C1 in all but the CC group, which is taken to indicate that the sounds suppress the initial visual V1 response in CC.

This is an impressive and well-controlled study, that extends our knowledge about the still little understood cortical processes behind sight-recovery in congenitally blind humans. The reported results are surely of great interest to a wide readership of the Journal. The reported experiment is simple but thoughtfully designed. The authors cleverly compare upper versus lower VF stimulations which allows them to demarcate the C1 as it reverses its polarity. Data analysis up to the standards in the field. While I could not spot any major issue, I have a few points that may be worth considering. Those are detailed below.

(1) The work reported here aims at assessing the stage of earliest visual cortical processing, hence the focus on the C1 component. The interval of analysis encompasses 50-100ms which clearly gauges the C1 component. But, as the authors may know, there is work showing that only the very initial part of the C1 50-70ms (C1e) reflects the V1 response exclusively (cf. Foxe & Simpson, 2002, EBR, Kelly et al. 2008, CerebCort). Around 90-100ms activity will propagate to early visual areas V2/V3. The upper-minus-lower VF difference ($\Delta C1$) will be helpful, as it cancels parts of the contribution from extrastriate areas. But this will not entirely eliminate such contributions because upper and lower VF representations in particular in V3 will differ in topography such that polarity reversals can be expected. It may, therefore be worthwhile checking just the very initial part of the C1.

(2) p. 8/ and Figure 2. The authors argue that the CC group does not show any C1 modulation in the audiovisual (AV) conditions. It appears however, that the incongruent AV condition shows a small effect that is similar to that in the DC group. The effect may not be significant, but it is clearly visible. Then, a significant difference between congruent AV versus visual only (V) is discussed, but what about the incongruent AV versus V? Shouldn't this comparison yield a significant difference to justify the conclusion that there is no C1 effect in this condition. The absence of the C1 in the congruent AV is indisputable, and overall, the authors' conclusions regarding the cross-modal suppression of V1 activity are justified. I still wonder whether a small C1 effect in the incongruent AV condition could speak for some degree of retinotopic specificity of the suppressive auditory projection to V1 in CC.

(3) A number of possible explanations for cross-modal influences on V1 activity are provided and discussed in large detail. The general tenet is to account the findings in terms of an atypical cross-modal brain network, that develops in a sensitive period after birth. The authors emphasize that congenitally blind humans show more 'extensive non-visual activity in the visual cortex'. In other words, the 'exuberant cross-modal' input would be expected to increase activity in V1. The authors

however observe that cross-modal stimulation suppresses the visual response in V1. What remains unaddressed in the discussion is why we see a suppression instead of enhancement. The authors favor an interpretation in terms of higher-level thalamic input. But those are unlikely to be inhibitory in nature. Pulvinar inactivation silences V1 activity. Coming back to point (2), why couldn't just simple competition between the auditory and visual representation in V1 account for the suppression effect? Of course, for this to be addressed further, it would be desirable to have an auditory-only stimulation to see whether this would elicit a lateralized response in V1. I do not expect the authors to provide those data in a revision, I just wonder how they envision the mechanism behind visual suppression in the CC group.

Reviewer #2 (Remarks to the Author):

The authors compared the difference in upper/lower visual field C1 ERPs for visual and congruent/incongruent auditory stimuli across groups, most notably participants with congenital cataract reversal. They found that the C1 difference wave was reduced during the concurrent stimuli trials only in the CC group, suggesting a multimodal effect of auditory stimulation on visual responses that was specific to these individuals. However, there were no behavioural differences between the groups.

This is a compelling manuscript with clear presentation of the methods and results. I appreciate the documentation of pilot data which informed some of the analysis decisions, and commend the presentation of individual data points.

I have a few points for clarification:

1) The main claim of the findings is that 'stimulus driven visual cortical activity (< 100 ms) was suppressed in sight-recovery individuals when concomitant sounds accompanied visual stimulation'. This is based on the reduction in the C1 difference wave between the visual only condition and the visual and auditory conditions. Of course, as you are using a difference value, the lack of C1 difference does not necessarily mean that the visual activation and accompanying C1 was suppressed, only that the subtraction of lower-upper visual stimulation came to zero (or close to). To support your claims about visual suppression, please could you include an ERP plot in the supplementary for the C1 waves for upper and lower visual fields separately, at least for the CC group and a control, for visual and visual/auditory. This will help to emphasise the source of the variation in the difference values, and hopefully demonstrate the classic C1 negativity for upper stimuli and positivity for lower stimuli. It is interesting that in the AVc condition, the CC topoplot is more negative over visual electrodes (although perhaps only moderately so).

2) Are the topographic plots presented in figure 2 representing the re-mapped data, corresponding to the right visual field stimuli, or is this the original data?

3) In figure S5.2, the topography of response looks markedly different in the time window 90-100ms, with a difference arising between hemispheres rather than in a centralised peak as before. Based on this pilot data, why did you then decide to use the whole window from 50-100ms?

Point-By-Point Replies to the Reviewers For
Sound Suppresses Earliest Visual Cortical Processing After Sight Recovery in
Congenitally Blind Humans

(Manuscript ID: COMMSBIO-23-1206-A)

Reviewers' comments:

Reviewer #1 (Remarks to the Author):

This study uses visual ERPs to investigate whether and how the well-documented effect of cross-modal (auditory) activation in early visual cortex (V1) of congenitally blind humans (bilateral congenital cataract, CC group) changes as a function of sight-recovery after surgery. The effect is compared to a group of participants with surgically reversed developmental cataract (DC group), as well as normal-sighted control groups, matching the cataract-groups (MCC, MDC). To assess cross-modal activation in early visual cortex, ERPs elicited by gratings presented at a fixed position in one of the four visual quadrants are analyzed. The positions are chosen to optimally derive the C1 component - a modulation of the VEP known to reflect activity generated in primary visual cortex. Cross-modal activation in V1 is assessed by co-presenting sounds with the gratings either from a congruent or incongruent spatial direction (sound appearing at the side of, or opposite to the grating). It is reported that unimodal visual stimulation elicits a clear C1 response in all groups. Cross-modal stimulation elicits a C1 in all but the CC group, which is taken to indicate that the sounds suppress the initial visual V1 response in CC.

This is an impressive and well-controlled study, that extends our knowledge about the still little understood cortical processes behind sight-recovery in congenitally blind humans. The reported results are surely of great interest to a wide readership of the Journal. The reported experiment is simple but thoughtfully designed. The authors cleverly compare upper versus lower VF stimulations which allows them to demarcate the C1 as it reverses its polarity. Data analysis up to the standards in the field. While I could not spot any major issue, I have a few points that may be worth considering. Those are detailed below.

(1) The work reported here aims at assessing the stage of earliest visual cortical processing, hence the focus on the C1 component. The interval of analysis encompasses 50-100ms which clearly gauges the C1 component. But, as the authors may know, there is work showing that only the very initial part of the C1 50-70ms (C1e) reflects the V1 response exclusively (cf. Foxe & Simpson, 2002, EBR, Kelly et al. 2008, CerebCort). Around 90-100ms activity will propagate to early visual areas V2/V3. The upper-minus-lower VF difference ($\Delta C1$) will be helpful, as it cancels parts of the contribution from extrastriate areas. But this will not entirely eliminate such contributions because upper and lower VF representations in particular in V3 will differ in topography such that polarity reversals can be expected. It may, therefore be worthwhile checking just the very initial part of the C1.

Response: The Foxe and Simpson (2008) study is of fundamental importance, indicating that numerous overlapping, non-retinotopic processes might mask V1 activity (or more generally, early visual cortical activity) in the C1 latency range. Of critical importance is that Foxe and Simpson (2008) presented stimuli on the horizontal meridian, and did not calculate the $\Delta C1$ as a difference wave between visual ERPs elicited by upper vs. lower visual field stimulations (UVF/LVF) as has been recommended for factoring out non-retinotopic activity¹. Similarly,

while Kelly et al. (2008) employed stimuli in UVF as well as LVF locations, the study also did not explicitly derive the $\Delta C1$ as a marker of early, retinotopic processing². Without an LVF – UVF subtraction, the individual visual ERP waves are confounded by non-retinotopic visual as well as common nonvisual neural activity, and indeed only the earliest part of the C1 wave can be reliably employed as a marker of retinotopic processing in V1. However, narrower time windows come with the risk that due to individual latency differences, noisier estimates of the C1 might be obtained. Calculating the $\Delta C1$ as in the present study, in contrast, allowed us to derive a marker of visual processing that is retinotopic by definition; the source analysis results included in the revised manuscript additionally indicate that the loci of $\Delta C1$ are consistent with an early visual cortical origin, with V1 being a major contributor (Fig. 3, p. 10 – 12).

While V1 might not be the sole generator of the $\Delta C1$ as operationalized in the present study, direct electrophysiological recording in non-human primates as well as topological considerations have indicated that it is likely to be the strongest contributor. In macaques, early V1 activity has been reported to be about 6 times stronger than V2 activity³. Additionally, an influential paper by Kelly et al. (2013) demonstrated strong topological constraints imposed by the structure of the early visual cortical surface for the generation of the C1 wave. Under these constraints, the systematic changes in C1 topography, observed for a shift of a visual stimulus within a visual field quadrant, cannot be explained by sources in areas V2/V3. In contrast, sources in area V1 can satisfactorily explain the within-quadrant topographical shifts⁴.

In the revised manuscript, while we have emphasized that the V1 is likely the major contributor of $\Delta C1$, we have carefully avoided any assertion that might be equivalent to claiming that the $\Delta C1$ is of a purely striate cortical origin. For example, in p. 4 we have changed the text as follows:

“This procedure allowed the separation of retinotopic activity from any unspecific activity (e.g., non-retinotopic neural activity as well as neural activities that are common to both the UVF and the LVF stimuli), and thus indexes the genuine retinotopic activity in early visual cortex, with V1 likely the strongest contributor^{4,5}. Any difference between the $\Delta C1$ between unimodal visual vs. crossmodal (audiovisual) conditions thus reflects sound evoked changes in bottom-up activity in early visual cortex, especially V1³⁷.”

Regardless of the exact relative contribution of the areas V1/V2/V3 to the $\Delta C1$, our results provide strong evidence that the earliest retinotopic visual cortical activity is suppressed by auditory information in individuals with reversed congenital cataracts, with a graded effect of spatial congruence (see following section).

(2) p. 8/ and Figure 2. The authors argue that the CC group does not show any C1 modulation in the audiovisual (AV) conditions. It appears however, that the incongruent AV condition shows a small effect that is similar to that in the DC group. The effect may not be significant, but it is clearly visible. Then, a significant difference between congruent AV versus visual only (V) is discussed, but what about the incongruent AV versus V? Shouldn't this comparison yield a significant difference to justify the conclusion that there is no C1 effect in this condition. The absence of the C1 in the congruent AV is indisputable, and overall, the authors' conclusions regarding the cross-modal suppression of V1 activity are justified. I still wonder whether a small C1 effect in the incongruent AV condition could speak for some degree of retinotopic specificity of the suppressive auditory projection to V1 in CC.

Response: We took advantage of the Bayesian analysis paradigm to investigate whether the AV_i condition's $\Delta C1$ might have been in-between that of the V and the AV_c condition. We found

that this indeed seems to be the case, indicating a degree of spatial specificity for the suppression of $\Delta C1$ by concurrent auditory stimulation. In the revised manuscript, the following part has been added to the results section (p. 10), noting that this analysis was exploratory.

“In the CC group, qualitatively, both at the grand average level as well as for individual data points, the $\Delta C1$ suppression appeared to be stronger for the spatially congruent (AV_c) than for the spatially incongruent audiovisual condition (AV_i ; see Fig. 2a – d). To examine whether the $\Delta C1$ suppression depends on spatial congruence, we modeled the $\Delta C1$ in the CC group in an exploratory analysis as a function of stimulus condition. An ordered factor with the levels (V , AV_i , AV_c), increasing in spatially specific crossmodal influence, served as the independent variable. The Bayesian analysis indicated substantial evidence for a general linear trend across the five electrodes in the CC group ($V > AV_i > AV_c$; $BF_{10} = 211.03$, positive ROPE test), indicating a $\Delta C1$ suppression modulated by spatial congruence.”

(3) A number of possible explanations for cross-modal influences on V1 activity are provided and discussed in large detail. The general tenet is to account the findings in terms of an atypical cross-modal brain network, that develops in a sensitive period after birth. The authors emphasize that congenitally blind humans show more ‘extensive non-visual activity in the visual cortex’. In other words, the ‘exuberant cross-modal’ input would be expected to increase activity in V1. The authors however observe that cross-modal stimulation suppresses the visual response in V1. What remains unaddressed in the discussion is why we see a suppression instead of enhancement. The authors favor an interpretation in terms of higher-level thalamic input. But those are unlikely to be inhibitory in nature. Pulvinar inactivation silences V1 activity. Coming back to point (2), why couldn’t just simple competition between the auditory and visual representation in V1 account for the suppression effect? Of course, for this to be addressed further, it would be desirable to have an auditory-only stimulation to see whether this would elicit a lateralized response in V1. I do not expect the authors to provide those data in a revision, I just wonder how they envision the mechanism behind visual suppression in the CC group.

Response: Previous work employing MRI in humans with reversed congenital cataracts have reported that audiovisual stimulation suppressed visual cortical processing in a crossmodal speech processing paradigm⁶. Of particular interest is that in this study visual cortical activity in the auditory-only condition was indistinguishable in the visual cortices of sight-recovery individuals and typically sighted controls. Other fMRI studies involving auditory stimuli in sight-recovery individuals with a history of congenital cataracts have either reported no evidence for an auditory activation of the visual cortex at group level⁷, or weak extrastriate cortical activation⁸. Thus, the fMRI evidence suggested that we would unlikely expect a simple non-retinotopic auditory enhancement of the visual cortical activity in individuals with reversed congenital cataracts. Rather, we expected a competition between auditory and visual inputs. As our hypothesis was confirmed we concluded that there might to some degree be a competition between sensory modalities during a sensitive period of brain development; the lack of visual input likely prevented a retraction of non-matching (auditory) inputs, which in the present study might have gated the $\Delta C1$ for concurrent audiovisual stimuli.

Research in murine and primate models have reported that the pulvinar can indirectly suppress V1 pyramidal neurons^{9,10}. Moreover, pulvinar inactivation silences V1, but focal activation has been found to enhance specific recipient locations in V1, while suppressing others¹⁰. We think that the property to gate, i.e., selectively enhance or suppress V1 activity makes the pulvinar

one of the suitable candidates as a node which might play a role in the $\Delta C1$ suppression observed in the present study.

We have clarified these speculations as possible mechanisms for our present results in the revised manuscript (p. 16).

Reviewer #2 (Remarks to the Author):

The authors compared the difference in upper/lower visual field C1 ERPs for visual and congruent/incongruent auditory stimuli across groups, most notably participants with congenital cataract reversal. They found that the C1 difference wave was reduced during the concurrent stimuli trials only in the CC group, suggesting a multimodal effect of auditory stimulation on visual responses that was specific to these individuals. However, there were no behavioural differences between the groups.

This is a compelling manuscript with clear presentation of the methods and results. I appreciate the documentation of pilot data which informed some of the analysis decisions, and commend the presentation of individual data points.

I have a few points for clarification:

1) The main claim of the findings is that ‘stimulus driven visual cortical activity (< 100 ms) was suppressed in sight-recovery individuals when concomitant sounds accompanied visual stimulation’. This is based on the reduction in the C1 difference wave between the visual only condition and the visual and auditory conditions. Of course, as you are using a difference value, the lack of C1 difference does not necessarily mean that the visual activation and accompanying C1 was suppressed, only that the subtraction of lower-upper visual stimulation came to zero (or close to). To support your claims about visual suppression, please could you include an ERP plot in the supplementary for the C1 waves for upper and lower visual fields separately, at least for the CC group and a control, for visual and visual/auditory. This will help to emphasise the source of the variation in the difference values, and hopefully demonstrate the classic C1 negativity for upper stimuli and positivity for lower stimuli. It is interesting that in the AVc condition, the CC topoplot is more negative over visual electrodes (although perhaps only moderately so).

Response: In the revised manuscript, we provided the C1 waves for the upper vs. lower visual field stimulations (UVF/LVF) separately for completeness (Supplementary Materials: *S4. C1 Wave Time Courses for Upper and Lower Visual Field Stimulation*). There are two potential confounds related to interpreting separate C1 wave time courses instead of their difference wave, i.e., the $\Delta C1^{1,11}$, which we briefly outline here as well as in the supplementary materials.

The first confound is that for the audiovisual conditions (AV_i and AV_c), the visual event-related activity is accompanied with auditory event-related activity, specifically the auditory P1-N1 complex¹², which is often larger on the scalp than the C1 wave itself. This confound would have made interpretation of separate ERPs waves to UVF and LVF stimulation difficult if not impossible, since due to component overlap, visual activity is masked to a large extent. To sidestep this confound, for audiovisual conditions we subtracted the unimodal auditory event-related potential (AERP) for plotting. For the audiovisual congruent conditions (AV_c), to derive the AERP, the electrodes in all EEG epochs with unimodal left auditory field stimulation were remapped and averaged together with the EEG epochs generated from unimodal right auditory field stimulation, as for the visual stimulation (see *Method*), serving as an estimate for the ipsilateral (right) auditory stimulus related activity. Thus, the plots depict ($AV_c, UVF - A_R$) and

($AV_{c, LVF} - A_R$) instead of ERPs to $AV_{c, UVF}$ and $AV_{c, LVF}$. Similarly, for plotting the C1 waves in the audiovisual incongruent conditions (AV_i), we subtracted the AERP estimate of the contralateral auditory stimulation (A_L), derived by remapping the electrodes of all unimodal right auditory field stimulation condition before averaging them together with EEG epochs generated from the left visual field stimulation.

The second confound is that early visual or audiovisual *non-retinotopic* activity is not accounted for in separate C1 wave plots for the UVF and LVF stimulation. Early non-retinotopic visual activity (e.g., the early P1 wave) onset has been reported to be only 10-20 ms after C1 wave onsets, making the $\Delta C1$ necessary for isolating feedforward retinotopic visual activity¹. Moreover, overlapping non-retinotopic audiovisual activity might lead to an overall shift of both waves for the audiovisual conditions.

Despite these caveats, the pattern of activity captured by the $\Delta C1$ was faithfully reflected in the separate C1 wave plots for UVF vs. LVF stimulation as well: In all groups and conditions, we observed C1 waves with canonical appearance except in the CC group for the audiovisual conditions. For the AV_i condition, partially overlapping C1 waves were qualitatively observed; for the AV_c condition, we observed an initial reversal of the expected C1 direction followed by overlapping C1 waves for UVF vs. LVF conditions. However, the effect sizes appear too small for a formal statistical analysis.

In the revised manuscript, the abstract was modified in addition to reflect that the present study investigated early *retinotopic* visual cortical activity: (stimulus driven retinotopic visual cortical activity (< 100 ms) was suppressed in sight-recovery individuals when concomitant sounds accompanied visual stimulation")

2) Are the topographic plots presented in figure 2 representing the re-mapped data, corresponding to the right visual field stimuli, or is this the original data?

Response: All plots including Fig. 2 represent remapped data, i.e., visual stimuli can be assumed to arrive from the right (= ipsilateral) side. We included this information in the result section (p. 6, no change undertaken)

3) In figure S5.2, the topography of response looks markedly different in the time window 90-100ms, with a difference arising between hemispheres rather than in a centralised peak as before. Based on this pilot data, why did you then decide to use the whole window from 50-100ms?

Response: Previous studies with similar paradigms have often employed 50 – 100 ms as the C1 latency range, which formed the basis of the pilot study^{2,11} as well.

From the $\Delta C1$ plots of the pilot study (Fig. S6.1) it appeared that the $\Delta C1$ was small, and by 90 ms, goes below zero. We think that the difference between hemispheres observed between 90 – 100 ms in the pilot study might have arisen because of this.

As outlined in S6.3: *Design/Analysis Decisions for Following Studies*, we reasoned that the smaller $\Delta C1$ waves observed in the pilot study might have been caused by two factors:

- In the pilot study, the participants sat in a brightly lit room, thus the pattern onset stimuli might have had less contrast compared to the stimuli in the main study.
- The mean interstimulus interval in the pilot study was 1.1 s, lower than in the subsequent studies, which might have led to a reduction of the $\Delta C1$ via refractory effects¹³, and might also have decreased its duration.

Based on these results we tested participants in the following studies in a dimly lit room and increased the mean interstimulus interval to 1.85 s as well (S6.3). In a subsequent, independent study, we found that the $\Delta C1$ waves appeared with typical latency, and had higher amplitude (see Fig. 1, next page)¹⁴, indicating that the stimulus changes made 50 – 100 ms an appropriate window for analyzing the $\Delta C1$. In the present study, the figures 2e and S4.1 similarly indicate that 50 – 100 ms is an appropriate analysis time range for the C1 wave. Thus, while it can be debated whether 50 – 100 ms was the optimal window for analyzing the $\Delta C1$ in the pilot study, the changes made to the visual stimuli based on this pilot study were successful in generating $\Delta C1$ waves with typical latencies in subsequent experiments. Based on the known spatiotemporal dynamics of early visual cortical activity, the latency range used in earlier studies^{2,11}, as well as the $\Delta C1$ wave latency in the present study which conformed to them, we employed the typical 50 – 100 ms in our analysis.

Figure 1: a. C1 Waves of typically sighted control groups in an earlier independent published report (Sourav et al., 2018), which used the same visual stimulus characteristics as employed in the present study¹⁴. MCC and MDC: typically sighted matched controls, respectively for the CC and the DC group, in the independent experiment. Error bands represent the standard error of the mean. Note the appropriateness of a 50 – 100 ms time window for the analysis of the C1 wave. b. Topography of the $\Delta C1$ (i.e., difference between ERPs to the LVF and the UVF stimuli) between 50–100 ms poststimulus. A canonical centralized topography for the $\Delta C1$ was observed as in the present study.

Additional Improvement to the Manuscript for Statistical Reproducibility

The present study employed Bayesian methods using the *brms* package in the R programming language^{15,16}. Brms uses a Hamiltonian Monte-Carlo sampler to draw samples from a distribution. Thus, the statistical estimates and Bayes factors can vary in different runs, which creates a problem for the reproducibility of the analysis.

In the revised manuscript, we ran each Bayesian model to draw 40,000 samples as recommended for robust Bayes factor estimation by the brms package. In addition, we set a fixed seed for the random number generator functions with the `set.seed()` function to guarantee reproducibility. In the revised manuscript, the variable estimates and Bayes factors have changed as a result. These changes are minor and involved no change of decision boundaries. In case of variable estimates, the changes were no more than $\pm 0.02 \mu\text{V}$.

References

1. Qu, Z. & Ding, Y. Identifying and removing overlaps from adjacent components is important in investigations of C1 modulation by attention. *Cognitive Neuroscience* **9**, 64–66 (2018).
2. Kelly, S. P., Gomez-Ramirez, M. & Foxe, J. J. Spatial Attention Modulates Initial Afferent Activity in Human Primary Visual Cortex. *Cerebral Cortex* **18**, 2629–2636 (2008).
3. Schroeder, C. E., Mehta, A. D. & Givre, S. J. A spatiotemporal profile of visual system activation revealed by current source density analysis in the awake macaque. *Cerebral Cortex* **8**, 575–592 (1998).
4. Kelly, S. P., Vanegas, M. I., Schroeder, C. E. & Lalor, E. C. The cruciform model of striate generation of the early VEP, re-illustrated, not revoked: A reply to Ales et al. (2013). *NeuroImage* (2013) doi:10.1016/j.neuroimage.2013.05.112.
5. Qu, Z. & Ding, Y. Identifying and removing overlaps from adjacent components is important in investigations of C1 modulation by attention. *Cognitive Neuroscience* **9**, 64–66 (2018).
6. Guerreiro, M. J. S., Putzar, L. & Röder, B. The effect of early visual deprivation on the neural bases of multisensory processing. *Brain* **138**, 1499–1504 (2015).
7. Guerreiro, M. J. S., Putzar, L. & Röder, B. The effect of early visual deprivation on the neural bases of auditory processing. *Journal of Neuroscience* **36**, 1620–1630 (2016).
8. Collignon, O. *et al.* Long-Lasting Crossmodal Cortical Reorganization Triggered by Brief Postnatal Visual Deprivation. *Current Biology* **25**, 2379–2383 (2015).
9. Fang, Q. *et al.* A Differential Circuit via Retino-Colliculo-Pulvinar Pathway Enhances Feature Selectivity in Visual Cortex through Surround Suppression. *Neuron* **105**, 355–369.e6 (2020).

10. Purushothaman, G., Marion, R., Li, K. & Casagrande, V. A. Gating and control of primary visual cortex by pulvinar. *Nature Neuroscience* **15**, 905–912 (2012).
11. Miller, C. E., Shapiro, K. L. & Luck, S. J. Electrophysiological measurement of the effect of inter-stimulus competition on early cortical stages of human vision. *NeuroImage* **105**, 229–37 (2015).
12. Sysoeva, O. V., Molholm, S., Djukic, A., Frey, H.-P. & Foxe, J. J. Atypical processing of tones and phonemes in Rett Syndrome as biomarkers of disease progression. *Transl Psychiatry* **10**, 1–12 (2020).
13. Coch, D., Skendzel, W. & Neville, H. J. Auditory and visual refractory period effects in children and adults: An ERP study. *Clinical Neurophysiology* **116**, 2184–2203 (2005).
14. Sourav, S., Bottari, D., Kekunnaya, R. & Röder, B. Evidence of a retinotopic organization of early visual cortex but impaired extrastriate processing in sight recovery individuals. *Journal of Vision* **18**, 22 (2018).
15. Bürkner, P. C. brms: An R Package for Bayesian Multilevel Models Using Stan. *Journal of statistical software* **80**, 1–28 (2017).
16. R Core Team. R: A language and environment for statistical computing. (2016).

REVIEWERS' COMMENTS:

Reviewer #1 (Remarks to the Author):

In this revision the authors have addressed the issues I had in a satisfying manner. I have no further comments or suggestions.

Reviewer #2 (Remarks to the Author):

Dear authors,

Thank you for your detailed and considered responses to the reviewer comments.
I am satisfied with the improvements to the manuscript and recommend publication.